# Examining the Influence of Omega-3 Fatty Acids on Performance, Recovery, and Injury Management for Health Optimization: A Systematic Review Focused on Military Service Members

**DOI:** 10.3390/nu17020307

**Published:** 2025-01-16

**Authors:** Melissa Rittenhouse, Saachi Khurana, Stephen Scholl, Christopher Emerson

**Affiliations:** 1Department of Military and Emergency Medicine, F. Edward Hébert School of Medicine, Uniformed Services University, Bethesda, MD 20814, USA; 2Henry M. Jackson Foundation for the Advancement of Military Medicine, Inc., Bethesda, MD 20817, USA

**Keywords:** military, omega-3 fatty acids, omega-3 index, performance, injury, recovery

## Abstract

Background/Objectives: Omega-3 fatty acids (*n*-3), recognized for their anti-inflammatory and brain health benefits, are being studied to enhance cognitive function, aid physical recovery, and reduce injury rates among military service members (SMs). Given the unique demands faced by this tactical population, this systematic review aims to evaluate the evidence of *n*-3 to support physical and mental resilience and overall performance. Methods: This review was conducted in accordance with Preferred Reporting Items for Systematic reviews and Meta-Analyses (PRISMA) guidelines and includes articles that assessed *n*-3 status or implemented *n*-3 interventions in relation to physical and cognitive performance, recovery, and injury outcomes (2006 to 2024). Of the 1606 articles yielded in screening through Covidence, 755 were irrelevant, leaving 226 studies for full-text eligibility. Of those 226 studies, 165 studies were excluded, and 61 studies were included in this review. Results: The results highlighted evidence-based findings in five key areas where omega-3 fatty acids are being evaluated to benefit military service members. These key areas include cardiopulmonary function, exercise recovery, cognitive function, injury recovery, and strength and power. While existing research suggests promising benefits, the most significant evidence was seen with cardiopulmonary function, exercise recovery, and cognitive function. Conclusions: Current research is promising and shows potential benefits, but the results are inconclusive and inconsistent. Future research is needed to determine optimal *n*-3 status, dose, and possibly type of *n*-3 across the various performance outcomes. Understanding these gaps in research will be essential to creating evidence-based *n*-3 guidelines for optimal performance of SMs.

## 1. Introduction

Military service members (SMs) face distinctive physical and mental challenges that define them as a specialized, tactical population. The combination of strenuous physical activity, sustained mental focus, and operational stress requires high resilience, making optimal health and performance a priority for military readiness. This level of demand often leads to increased wear on the body and mind, making it necessary to explore nutritional strategies that can support both physical and cognitive function. Notably, omega-3 fatty acids (*n*-3) have gained attention due to their supportive role in cardiovascular function [1,2], cognitive function [3], injury prevention [4] and treatment [5], and recovery from strenuous exercise [6]. All of these health outcomes are crucial to maintaining peak performance under demanding situations.

*N*-3—specifically eicosapentaenoic acid (EPA) and docosahexaenoic acid (DHA)—are long-chain fatty acids predominantly derived from marine foods such as fish and algae, with salmon, herring, mackerel, and sardines offering the highest concentrations [7]. Alpha-linolenic acid (ALA), another form of *n*-3 found in plant sources such as chia seeds, flaxseeds, and walnuts, can be partially converted to EPA and DHA in the body, although this conversion is relatively limited, with a conversion rate from ALA to EPA/DHA of 5% [7,8].

One of the most common markers used to assess *n*-3 status is the Omega-3 Index (O3I), a blood biomarker representing the sum of EPA and DHA as the percent total of erythrocyte fatty acids [7]. Higher O3I levels are associated with reduced risk for cardiovascular disease and improved overall health outcomes [7], with an O3I of 8% [9] as the optimal standard. Although diet remains the primary source of EPA and DHA, supplementation has potential to raise O3I levels in individuals who cannot meet their needs through food alone [9].

The average *n*-3 status in many populations, including SMs, is suboptimal. Research indicates that 68% of American adults, including SMs, fall short of the recommended two to three servings of oily fish per week and have low *n*-3 levels [10]. In terms of health outcomes, 89% of adults with low O3I levels (<4%) fall into the high cardiovascular risk category [10]. Similarly, SMs that fall below an O3I of 4% are also at high risk for chronic disease [10]. This problem of low O3I prompts questions about whether an increase in *n*-3 intake and improved *n*-3 status could bridge the gap and support optimal health and performance outcomes for tactical populations, including SMs. The objective of this review is to evaluate the role of *n*-3 in enhancing performance, recovery, and injury prevention that could enhance both the physical and mental resilience of SMs.

## 2. Materials and Methods

The current systematic review follows the Preferred Reporting Items for Systematic Reviews and Meta-Analyses (PRISMA) guidelines [11]. The research team consisted of a nutrition researcher, a research coordinator, and a medical student. During the review and selection process, the research coordinator and medical student independently screened the articles. Any disagreements were resolved by the nutrition researcher at each stage in the PRISMA flow chart, using the specified inclusion and exclusion criteria (Figure 1).

All original articles included in this review were published in English between 2006 and 2024. They included randomized control trials and cross-sectional, cohort, observational, quasi-experimental studies. Systematic and narrative reviews were excluded. The study inclusion criteria were active or healthy participants aged 18–65 who received *n*-3 supplementation or a measurement of *n*-3 status, with performance outcomes. These outcomes encompassed measures of strength and power, cardiopulmonary function, cognitive performance, injury prevention, and exercise recovery. Exclusion criteria were no *n*-3 intervention or status, elderly population, and/or no performance outcomes.

### 2.1. Literature Search Strategy

Covidence, a systematic review tool, was used to search abstracts and titles for keywords following PRISMA 2020 guidelines [11,12]. The search strategy was completed in October 2024, using four online databases (PubMed, Embase, Ovid, and CINAHL) and the following keywords: military, active duty, soldier, *n*-3, omega-3 status, omega-3 index, omega-3 supplementation, injury, performance, cognitive function, delayed onset muscle soreness, recovery, and physical fitness.

The initial literature search yielded 1606 potential articles for screening. After identifying 618 duplicates, 988 articles remained, which were screened and included based on title and abstract. Of those, 226 met the criteria for full-text articles to be reviewed, and of those, 165 were excluded for the following reasons: wrong study design, wrong intervention, wrong patient population, wrong outcomes, or systematic review. In total, 61 full-text articles met Population, Intervention, Comparison, and Outcomes (PICO) study design criteria.

### 2.2. Quality Assessment

The 2010 Effective Public Health Practice Project (EPHPP) Quality Assessment Tool was used to assess the studies for selection bias, study design, confounders, blinding, data collection methods, and withdrawals and drop-outs [13]. The EPHPP tool is an instrument for evaluating the methodological quality of quantitative studies, providing a comprehensive assessment across various domains to ensure the reliability and validity of study findings [13]. Of the 61 articles selected, only 3 received a “weak” score, while the remaining 58 (95%) received “moderate” or “strong” scores.

## 3. Results

This research examined the role of *n*-3 supplementation, intake, and/or status on performance outcomes. For consistency of reporting and comparing the results of the supplementation studies, we converted amounts, where necessary, to mg. All doses are reported as daily amounts; therefore, this is not notated throughout the results. In most cases, researchers provided a total amount of supplementation, as well as specific amounts of EPA and DHA. For these, we report amounts in parentheses, with the first number being the total amount, followed by EPA and then DHA.

Five key domains that assess the impact of *n*-3 on health and performance were created based on the results of the literature review:Cardiopulmonary function explores the influence of *n*-3 supplementation on heart rate, blood flow, pulmonary function, and cardiovascular fitness, emphasizing mechanisms that support exercise performance.The exercise recovery domain evaluates biomarkers associated with recovery, including inflammatory and oxidative stress markers, muscle damage, soreness, and range of motion.The cognitive function domain investigates outcomes related to attention, reactivity, information processing, executive functioning, and overall cognitive health.Injury recovery assesses the role of *n*-3 in promoting prevention and recovery from traumatic brain injuries (TBIs) and musculoskeletal injuries (MSKIs).The strength and power domain specifically focuses on the influence of *n*-3 on associated performance outcomes. Together, these findings provide a comprehensive understanding of the potential benefits of *n*-3 across these critical areas.

### 3.1. Cardiopulmonary Function

*N*-3, particularly with EPA and DHA, has been explored for its potential to enhance cardiovascular and pulmonary function. The following studies have investigated its effects on heart rate (HR), blood pressure (BP), pulmonary function, and measures of cardiovascular fitness such as VO2max. This section summarizes the beneficial and null results reported for this domain.

#### 3.1.1. Heart Rates (Max, Peak, Variability, and Resting) and BP

Several studies highlight the potential benefits of *n*-3 on heart rate (HR), heart rate variability (HRV), and blood pressure (BP) [14,15,16,17,18,19,20]. A 2021 study found that DHA-rich fish oil (2000 mg, 140 mg + 560 mg) supplementation slowed resting HR during cardiovascular reflex challenges [14]. Similarly, another study observed a dose-dependent reduction in resting HR during stress tasks, with high doses of *n*-3 (3400 mg, 1581 mg + 1275 mg) having a more pronounced effect compared to lower doses (850 mg, 395 mg + 319 mg) [15]. A 2008 study reported that 3200 mg of *n*-3 (800 mg DHA + 2400 mg EPA) reduced HR, peak HR during exercise to exhaustion, and steady-state submaximal HR, while also enhancing myocardial oxygen efficiency [16]. Another 2008 study found that 6000 mg of fish oil (360 mg EPA + 1560 mg DHA) for 12 weeks reduced HR both at rest and during submaximal exercise and improved HRV compared to placebo [17]. Further evidence demonstrated that 6000 mg of DHA-rich fish oil (360 mg EPA + 1560 mg DHA) decreased HR and diastolic BP during exercise [18]. In terms of HRV, one study identified higher O3I levels correlated with faster post-exercise HR recovery, especially in women, and another noted improved HR recovery time after exercise with lower-dose fish oil supplementation (140 mg EPA + 560 mg DHA) [19,20]. Notably, one study found that a high dose of *n*-3 (3400 mg, 1581 mg + 1275 mg) significantly reduced both HR and BP, with the magnitude of reduction remaining consistent between rest and stress conditions. The BP reduction was primarily driven by decreases in stroke volume and cardiac output. However, a low dose (850 mg, 395 mg + 319 mg) reduced HR relative to placebo, but had no effect on BP [15].

In contrast, some studies reported no significant effects of *n*-3 on HR, HRV, or BP. A 2014 study found that low-dose fish oil supplementation (560 mg DHA + 140 mg EPA) did not affect peak HR during intense exercise [20]. Similarly, another study noted no effect of *n*-3 supplementation (6000 mg, 3000 mg + 2000 mg) on resting or exercise HR [21]. Furthermore, another study observed no significant differences in HR or BP during exercise among EPA, DHA, or control groups [22]. Regarding HRV, one study reported no significant changes during cardiovascular reflex challenges, despite increased O3I levels [14]. For blood pressure, one study found that higher serum long-chain *n*-3 PUFA concentrations were associated with lower resting HR but had no effect on peak HR, HR recovery, or BP [23].

#### 3.1.2. Blood (CO, SV, and SVR)

*N*-3 supplementation demonstrated several positive effects on cardiovascular function in some studies [21,24]. One study found that a 6000 mg dose of supplemental *n*-3 (3000 mg + 2000 mg) increased stroke volume (SV) and cardiac output (CO) during moderate workloads while tending to attenuate decreases in systemic vascular resistance (SVR) [21]. The same supplementation also reduced resting mean arterial pressure (MAP) [21]. Another study reported *n*-3 supplementation of 1300 mg (660 mg EPA + 440 mg DHA) increased baseline nitric oxide (NO) levels, highlighting a potential mechanism for improved vascular function [24].

Despite these promising findings, other studies noted no significant benefits in specific cardiovascular outcomes. One study found no differences in CO during exercise between groups receiving *n*-3 supplementation (4700 mg EPA or DHA) and control. While SVR was lowered at 5 h and during exercise in the DHA group, this effect was not observed in the EPA group or the control group [22]. Additionally, another study observed no reduction in MAP during exercise following *n*-3 supplementation [21].

#### 3.1.3. Pulmonary Function (Forced Vital Capacity [FVC] and Forced Expiratory Volume [FEV1])

*N*-3 has shown positive effects on pulmonary function in some studies [25,26]. A study reported significant improvements in forced vital capacity (FVC), forced expiratory volume (FEV1), and other pulmonary markers in wrestlers who consumed 1000 mg of *n*-3 (180 mg EPA + 120 mg DHA) [26]. Similarly, other research found slight improvements in FVC, FEV1, and peak expiratory flow in individuals who were supplemented with a low dose of *n*-3 (330 mg, 91.5 mg + 63.0 mg) combined with vitamin D [25]. However, not all studies demonstrated improvements in pulmonary function with *n*-3 supplementation. One study, for example, found no significant differences in FVC or FEV1 between groups receiving high-dose *n*-3 (~5000 mg, 1000 mg + 3500 mg) and a placebo group [27].

#### 3.1.4. Cardiovascular Fitness

Some studies have highlighted the potential of *n*-3 to enhance aerobic capacity and exercise efficiency [24,28,29]. One reported an increase in VO2max following *n*-3 supplementation, while another study observed a significant rise in VO2peak among participants receiving 2234 mg EPA + 916 mg DHA [24,28]. The latter study also found a positive correlation between O3I and running economy [28]. In terms of exercise oxygen efficiency, a study reported reduced oxygen consumption during submaximal exercise with *n*-3 supplementation of 2000 mg (140 mg + 560 mg) [29].

Not all studies observed improvements in VO2max or related measures following *n*-3 supplementation. For example, one study found no significant changes in VO2peak among runners after supplementation (2234 mg EPA + 916 mg DHA), and another similarly reported no significant effects on VO2max with supplementation (3600 mg, 914 mg + 399 mg) [30,31].

### 3.2. Exercise Recovery

*N*-3, particularly EPA and DHA, shows promise in supporting exercise recovery by mitigating inflammation, reducing muscle damage, and alleviating delayed-onset muscle soreness (DOMS). This section specifically examines the impact of *n*-3 supplementation on various exercise recovery biomarkers, soreness, and range of motion (ROM).

#### 3.2.1. Exercise Recovery Biomarkers

Evidence suggests *n*-3 can influence key exercise recovery markers, including C-reactive protein (CRP) [32,33,34], interleukin 6 (IL-6) [34,35,36], and tumor necrosis factor-alpha (TNF-α) [35,37], which are inflammatory markers; creatine kinase (CK) [34,37,38,39], a muscle damage marker; and various oxidative stress markers [40,41,42,43].

#### 3.2.2. Inflammatory Markers (CRP, IL-6, TNF-α, etc.)

A study measuring the relationship between self-reported *n*-3 dietary supplement use and inflammation found that regular users of *n*-3 dietary supplements had significantly lower CRP levels before and after exercise as compared to non-users [32]. Another study found a statistically significant difference in CRP levels in subjects with a higher O3I at 24 h and a trend toward significance over 96 h [33]. DHA supplementation was found to reduce IL-6 levels by 32% in the acute recovery phase compared to placebo [34]. Similar research found that *n*-3 supplementation (1040 mg, 715 mg EPA + 286 mg DHA) effectively attenuated IL-6 increases post-exercise [35]. Additional research reported that *n*-3 supplementation (600 mg EPA + 260 mg DHA) significantly attenuated IL-6 increases post-exercise, suggesting a protective effect against exercise-induced inflammation [36].

A study with acute supplementation of DHA (2000 mg/day) showed modest reductions in CRP levels following eccentric exercise, but the results were not statistically significant [34]. Similarly, another study found that *n*-3 supplementation (1040 mg, 715 mg + 286 mg) had no significant effect on CRP [35]. Other studies also observed no significant changes in CRP levels following *n*-3 supplementation [44,45]. One reported that fish oil supplementation (4000 mg, 2000 mg + 800 mg) failed to alter CRP levels within 48 h post-cycling high-intensity interval training (HIIT), consistent with findings from the other, which noted no significant effects on CRP levels or group–time interactions with 4000 mg/day of DHA + EPA supplementation [44,45]. Many studies showed mixed results for CRP, with most showing no benefit. The opposite is true for IL-6: only one study reported no significant differences in IL-6 compared to the placebo [38].

Another marker of inflammation is TNF-α. One study demonstrated that green-lipped mussel oil (1200 mg/day) significantly reduced post-exercise TNF-α levels, but another showed no significant differences in TNF-α levels [35,37].

#### 3.2.3. Muscle Damage Markers

Several studies evaluated the effects of *n*-3 supplementation on muscle damage and recovery post-exercise, with varying results [33,34,35,37,38,39,41,45,46]. One study reported a significant reduction in CK levels post-exercise with supplementation (600 mg EPA + 260 mg DHA) [38]. Similarly, another observed that *n*-3 supplementation at 4000 mg significantly reduced CK levels after high-intensity interval training (HIIT), supporting improved recovery compared to placebo [44]. One study observed a statistically significant 12.5% reduction in CK levels with DHA supplementation (2000 mg) [34]. Another found that green-lipped mussel oil (~58 mg EPA + 44 mg DHA) significantly attenuated CK post-exercise, further highlighting the benefits of *n*-3 for reducing muscle damage markers [37]. Finally, a study assessing different amounts of *n*-3 observed the most significant reduction in CK levels in the 6000 mg group, compared to the placebo, 2000 mg, and 4000 mg groups post-exercise, further demonstrating decreased markers of muscle damage following *n*-3 supplementation [39].

A 2010 study found no differences in CK levels between *n*-3 supplementation (3000 mg of fish oil, 1300 mg + 300 mg) and placebo [41]. Kyriakidou et al. noted no differences in CK levels between supplementation groups due to exercise-induced muscle damage (EIMD) [35]. Again, another study noted no differences in CK levels between the supplementation and placebo groups following eccentric exercise [33]. A study assessing the impact of different types of *n*-3, reported that neither the EPA + DHA nor DHA groups mitigated an increase in CK levels [45]. Similarly, research found significant CK elevations from pre-exercise to post-exercise across all groups, reflecting muscle damage, and no significant differences between supplementation and placebo groups [46]. Overall, while some studies demonstrated the potential of *n*-3 supplementation to reduce CK levels and support recovery, others indicated variability in effects based on dose, composition, and timing. This highlights the need for further research to clarify *n*-3 efficacy in mitigating exercise-induced muscle damage.

#### 3.2.4. Oxidative Stress Markers

*N*-3 supplementation has also shown effects on various other oxidative stress markers [40,41,42,43]. A 2009 study found that *n*-3 supplementation (1300 mg fish oil, 30% EPA + 20% DHA), helped protect against oxidative stress during recovery following endurance exercise [40]. Similarly, a study reported that fish oil supplementation (3000 mg, 1300 mg + 300 mg) significantly reduced oxidative stress markers [41]. Further supporting these findings, one group demonstrated that re-esterified DHA supplementation (≥1050 mg) significantly reduced oxidative stress markers, showing a dose-dependent antioxidant effect [42]. Additionally, a different study found that krill oil supplementation (550 mg EPA + DHA and 150 mg choline) increased antioxidant capacity and significantly reduced oxidative stress markers compared to a placebo [43]. These results suggest that *n*-3, across various sources and dosages, may play an important role in mitigating oxidative stress and supporting antioxidant defenses during post-exercise recovery.

#### 3.2.5. Delayed-Onset Muscle Soreness (DOMS)

The effectiveness of *n*-3 in mitigating DOMS is variable, with some evidence of significant reductions in muscle soreness [33,35,36,37,38,39,44,45,47,48,49]. However, not all studies showed discernible benefit. One study found that EPA + DHA supplementation significantly reduced DOMS at 1 and 3 days post-exercise, while a later study by the same authors reported no differences in muscle soreness with similar supplementation [36,38]. Another study demonstrated that green-lipped mussel oil significantly reduced DOMS at 72 and 96 h post-exercise [37]. Similarly, other research reported significantly less DOMS at 1 and 2 days post-exercise with 600 mg EPA + 260 mg DHA [47]. A separate group found that individuals with a higher O3I reported less DOMS-related pain at 72 and 96 h post-exercise [33]. One particular study noted a strong dosing effect in the interaction between treatment and time for perceived soreness, with all groups (placebo, 2000 mg, 4000 mg, and 6000 mg) reporting the highest level of soreness at 24 h [39]. Additional research measured perceived pain in the anterior thigh, posterior thigh, and calf as a marker for DOMS and found significant reductions in calf pain scores, but not in the anterior or posterior thigh, following high-intensity interval training with *n*-3 supplementation compared to placebo, suggesting a partial reduction in HIIT-induced DOMS [44]. Black et al. found a moderate beneficial effect on muscle soreness and fatigue with 1546 mg of *n*-3 polyunsaturated fatty acid (551 mg EPA + 551 mg DHA) [48]. Similarly, other research also reported significantly lower muscle soreness in the *n*-3 group compared to placebo at 24 h post-exercise [35]. A 2011 study found that *n*-3 supplementation (3000 mg, 2000 mg + 1000 mg) reduced the magnitude of soreness increase by 15% compared to control [49]. In 2024, a study assessing different types of *n*-3 also reported significantly lower muscle soreness in DHA and EPA groups at 48 h compared to placebo, but not in the combined EPA + DHA group [45].

While there is some evidence to support a potential benefit of *n*-3 in reducing DOMS and perceived soreness, there are also studies that found no effect or significance. One study observed reductions in muscle soreness with fish oil (6000 mg), although these were not statistically significant [50]. Similarly, other research found no significant effects of *n*-3 supplementation on DOMS [41]. Likewise, DiLorenzo et al. found no significant differences in muscle soreness between a placebo and DHA supplementation (2000 mg) [34]. Furthermore, a study found no significant group–time interactions for muscle soreness with fish oil supplementation (3200 mg) [51]. Similarly, a study observed significant main effects for time, but not between supplementation groups [52]. Lastly, an alternate study found no significant differences in perceived soreness between the placebo or *n*-3 supplementation groups [46].

#### 3.2.6. Range of Motion (ROM)

A 2016 study reported significantly greater ROM when participants were provided 600 mg EPA + 260 mg DHA, compared to a placebo during the first 1–5 days post-exercise [36]. Similarly, a later study within the same lab observed that ROM was significantly higher in the EPA + DHA group immediately following eccentric contractions [38]. One study found that green-lipped mussel oil supplementation effectively preserved ROM and significantly reduced joint stiffness [37], yet another study demonstrated that higher plasma EPA levels were significantly associated with improved ROM post-exercise [47].

Some research showed no significant differences in ROM outcomes, indicating no significant differences in ROM between groups, with other similar research finding insignificant ROM differences between groups during either the acute recovery phase or the overall exercise period [34,45]. An additional study also noted no changes in hip and knee ROM or relative torque force (RTF) at any time point or between groups [46].

These findings suggest that the variability in outcomes across studies highlights the need for further research to identify optimal supplementation strategies affecting ROM improvements.

### 3.3. Cognitive Performance

Various forms of *N*-3 have been studied for their potential to enhance cognitive functions, including attention, reactivity, information processing, and executive function [53,54,55,56,57,58]. This section highlights findings from a select number of studies, emphasizing the effects of *n*-3 supplementation on these cognitive domains and overall cognitive performance.

#### 3.3.1. Attention and Reactivity

One study reported that after 21 days of supplementation with fish oil (1200 mg EPA + 600 mg DHA), athletes exhibited improved reaction times and increased vigor [54]. Similarly, a study of elite female soccer players found that DHA supplementation (3500 mg) led to significant improvements in complex reaction time, complex reaction efficiency, and neuromotor function [55]. Building on these findings, a similar study found that *n*-3 supplementation (2800 mg, 1600 mg + 800 mg) reduced reaction time in two different attention tests and improved mood, evidenced by reduced anxiety and increased vigor [53]. A cross-sectional study examined the relationship between *n*-3 status (categorized into low, middle, and high tertiles) and cognitive function, revealing that participants in the low O3I tertile scored the lowest in attention compared to those in the middle and high tertiles [56]. These studies suggest that *n*-3 can positively influence reactivity and attention, potentially through effects on the central nervous system.

#### 3.3.2. Information Processing and Executive Function

Regarding cognitive information processing, a study showed that a dietary intake of *n*-3 has been associated with improvements. Another cross-sectional study found that higher intake of *n*-3 was inversely associated with the risk of overall cognitive function impairment, particularly processing speed [57]. Research also observed that the high *n*-3 tertile group achieved the highest mean scores for information processing, although the differences among the low, middle, and high *n*-3 tertile groups were not statistically significant [56]. Overall, these findings support the theory that *n*-3 may enhance cognitive processing, particularly attention and memory.

Although some studies have suggested a positive effect of *n*-3 on executive function, the evidence is mixed. One particular study examined the impact of *n*-3 supplementation on cognitive performance of U.S. military officers and found no significant differences between treatment groups for executive function after 20 weeks of *n*-3 supplementation due to low compliance [58]. Another evaluated executive function based on low (<5.47%), medium (5.47–6.75%), and high (>6.75%) O3I results. Executive function scores were highest in the middle tertile [56].

#### 3.3.3. Overall Cognitive Function

Overall cognitive function has also been shown to benefit from *n*-3. One study found that higher *n*-3 intake, particularly from fatty fish, was associated with reduced risk of impaired cognitive function in middle-aged individuals [57]. In contrast, another study reported no significant improvements in cognitive function with supplemental *n*-3 [58].

### 3.4. Injury Recovery

#### 3.4.1. Traumatic Brain Injury

Growing evidence indicates that *n*-3 supplementation may help reduce the risk of traumatic brain injuries (TBIs) [59,60,61]. A 2021 study found that *n*-3 supplementation (2880 mg, 560 mg + 2000 mg) in American football players attenuated increases in neurofilament light chain (Nf-L) levels, a marker of neural damage, suggesting that *n*-3 may exhibit protective effects against sports-related brain injury [59]. In line with this, a similar 2022 study demonstrated neuroprotective trends on functional connectivity in American football players exposed to repetitive head impacts. Their study showed some preservation of white matter structure and reduced daytime sleepiness among players receiving DHA and EPA (2442 mg + 1020 mg), although depression and insomnia scores remained unaffected. The Nf-L levels at the end of supplementation did not differ between the supplementation and placebo groups [60]. These neuroimaging findings suggest that *n*-3 supplementation might mitigate damage from sub-concussive impacts, but further evidence is needed to validate these preliminary results.

Building on this evidence, another study examined the effects of different doses of DHA supplementation (2000 mg, 4000 mg, and 6000 mg) on serum Nf-L in American football players [61]. The study found that DHA likely attenuated serum Nf-L levels across all groups after training camp, a period associated with increases in Nf-L. During the season, when serum Nf-L levels were also elevated, DHA supplementation likely reduced the increase compared to the placebo. The 2000 mg group showed a consistent reduction in serum Nf-L levels compared to the placebo across all time points. For the 4000 mg and 6000 mg groups, the results were less clear, except at the end of the season, where the 4000 mg group demonstrated a 91% reduction. These findings highlight the potential role of *n*-3 in mitigating neural damage [61].

#### 3.4.2. Musculoskeletal Injury

In the realm of physical injury prevention, a study observed that recreational runners with higher O3I levels were less prone to running-related injuries (RRIs), with an elevated arachidonic acid (AA)/EPA ratio correlating with greater RRI incidence [62]. Similarly, another study also found that individuals with degenerative rotator cuff tears exhibited lower O3I levels compared to healthy controls [63].

Researchers have also evaluated *n*-3 intake and status as potential treatment and recovery approaches for injury. One study extended the benefits of *n*-3 supplementation to young women facing muscle disuse, as the *n*-3 group experienced significantly less muscle atrophy and higher muscle protein synthesis rates during immobilization, underscoring the role of *n*-3 in maintaining muscle integrity under inactivity [64]. Similarly, another study reported enhanced muscle endurance and strength following *n*-3 supplementation (2100 mg, 600 mg + 1500 mg) in individuals with spinal cord injuries [65]. Furthermore, a study evaluated *n*-3 supplementation of 285 mg/day (170 mg EPA + 115 mg DHA) in patients with rotator cuff-related shoulder pain, showing minor improvements in disability and pain outcomes, although effects were limited, suggesting that *n*-3 might offer modest benefits to joint health when combined with exercise and physical therapy [66].

### 3.5. Strength and Power

Research on *n*-3 demonstrates potential health benefits in general, but results regarding its effects on performance and neuromuscular adaptations remain inconsistent. Two studies identified positive associations between dietary *n*-3 intake and handgrip strength (HGS), highlighting the potential role of *n*-3 in supporting strength [59,67]. Additionally, one study found that erythrocyte EPA was predictive of plank performance after adjusting for body composition. Every 0.1% increase in EPA translated into a 5.4-point improvement in plank score [68]. A 2024 study found that jump performance, strength, and power metrics improved more rapidly in the EPA and DHA supplementation groups compared to placebo [45]. Similarly, in 2017 a study reported enhanced squat jump and countermovement jump performance after acute supplementation in the higher-EPA-dose group (750 mg EPA + 50 mg DHA) compared to both the lower-dose group (150 mg EPA + 100 mg DHA) and the control group [52]. In professional rugby players, a study reported that *n*-3 supplementation helped maintain explosive power during pre-season training [48]. Similarly, one study performed a Wingate test, a widely recognized method for assessing anaerobic power and fitness [69], and found that *n*-3 supplementation reduced Wingate percent power drop (4.76 ± 3.4%) in the *n*-3 group, compared to placebo [70]. However, findings were less consistent in a later study, where *n*-3 supplementation did not significantly impact neuromuscular adaptations during sprint interval training [71]. Similarly, an older study observed no significant impact on running performance or anaerobic performance in soccer players [72].

## 4. Discussion

### 4.1. Cardiopulmonary Function

While direct evidence linking *n*-3 to performance improvements is limited, research demonstrates promising improvements in cardiovascular markers, which may enhance performance outcomes. Studies have reported reductions in resting heart rate (HR) and in HR during submaximal and maximal steady-state exercise [14,15,16,17,18,20]. A lower HR indicates more efficient oxygen utilization by the heart, a key marker of cardiovascular fitness [73]. Additionally, other studies have observed faster HR recovery and higher HRV, reflecting greater adaptability to exercise and improved ability to recover from cardiovascular strain, which occurs during strenuous exercise [73]. This adaptability is essential in the military, where personnel often face demanding physical tasks followed by brief recovery periods, such as during combat operations or high-intensity interval training.

Some research observed that *n*-3 increased stroke volume (SV) and cardiac output (CO), indicating that the heart pumps more blood with each beat. Consequently, systemic vascular resistance (SVR) decreased, allowing for greater blood flow with less resistance, which enhances cardiovascular function during exercise [21]. Similarly, another study found that *n*-3 increased baseline nitric oxide (NO) levels, promoting blood vessel relaxation and improving blood flow [24]. This, in turn, helped lower blood pressure and enhance oxygen delivery, ultimately supporting improved exercise capacity [73]. These cardiovascular benefits are particularly useful to SMs who operate under challenging conditions, supporting physical readiness.

### 4.2. Exercise Recovery

The potential benefits of *n*-3 for exercise recovery hold significant implications for improving physical resilience and recovery from day-to-day demands for SMs. The observed reductions in inflammation, muscle damage, oxidative stress, and delayed-onset muscle soreness (DOMS) could contribute to enhanced recovery and overall performance, which are crucial for SM engaged in demanding physical activities.

For instance, reductions in inflammatory biomarkers such as CRP and IL-6 suggest that *n*-3 may help attenuate exercise-induced inflammation, a common challenge faced by SM during training or operational activities [32,34]. This anti-inflammatory effect may not only support quicker recovery but also may reduce the risk of injury by moderating the inflammatory response. Furthermore, the observed reductions in muscle damage markers, such as CK, indicate that *n*-3 could play a vital role in accelerating recovery after intense physical exertion [38,39]. Faster recovery from muscle damage could potentially improve training consistency and performance, ensuring SMs maintain peak physical conditioning, an essential component of military readiness.

Additionally, the positive impact of *n*-3 on oxidative stress may bolster antioxidant defenses, protecting against cellular damage induced by intense physical activity [40,43]. This effect could be particularly beneficial in countering the oxidative stress generated during high-intensity training and mission-specific tasks, helping SMs recover more efficiently and maintain optimal physiological function.

Finally, reported reductions in DOMS and improvements in ROM suggest that *n*-3 may help alleviate the discomfort associated with muscle soreness, improving mobility and flexibility in the post-exercise period [36,47]. This could be especially beneficial for SMs who need to maintain both mobility and agility during operational or training activities, where swift recovery from muscle soreness is crucial for operational success.

### 4.3. Cognitive Function

The positive effects of *n*-3 on cognitive functions such as attention, reactivity, and information processing are particularly relevant to SMs, given the cognitive demands and high-stress environments that are characteristic of military operations. For example, the findings of three studies demonstrated improved reaction times and neuromotor function following *n*-3 supplementation, which directly align with the needs of SMs who must maintain quick decision-making abilities and mental agility during high-pressure situations [53,54,55]. In military settings, where the ability to react rapidly and maintain focus is crucial for success, *n*-3 could play a significant role in enhancing these cognitive capacities. The association between higher *n*-3 intake and reduced cognitive impairment risk is also relevant for SMs, as maintaining cognitive health throughout a military career is vital for long-term performance and mental resilience [57].

The reported benefits of *n*-3 on information processing and attention also support the argument for its potential role in optimizing the cognitive abilities of SMs [56]. With the military placing high demands on attention and processing speed—whether in tactical operations, training exercises, or decision-making scenarios—improvement in these cognitive domains could lead to better performance outcomes.

### 4.4. Injury Recovery

The role of *n*-3 in injury prevention and recovery has garnered significant attention due to its anti-inflammatory and neuroprotective properties. This section explores key findings on *n*-3 supplementation in relation to TBI and MSKI.

Several studies demonstrated that high doses of EPA and DHA might provide neuroprotection, a finding of particular relevance for populations exposed to repetitive head impacts or high-risk environments. Two similar studies reported attenuation of neurofilament light chain (Nf-L) levels, a biomarker of neural damage, in athletes experiencing repetitive head impacts [59,61]. This finding is crucial for SMs, as they are frequently subjected to training and operational environments with elevated risks of traumatic brain injuries (TBIs) and concussions.

One of the two studies further demonstrated dose-dependent reductions in Nf-L levels, with DHA doses ranging from 2000 to 6000 mg during high-impact periods. These results suggest that higher doses of *n*-3, particularly DHA, may offer stronger protective effects against neural damage [61]. For SMs, such neuroprotective benefits could mitigate the long-term consequences of TBI, potentially preserving cognitive function and operational readiness. Alternatively, a different study utilized a moderate dose of 2442 mg DHA and 1020 mg EPA per day over seven months, observing preservation of white matter [60]. Improved white matter integrity could translate to better decision-making and sustained cognitive performance under operational stress, a crucial aspect of military effectiveness. Another study similarly supplemented with 2880 mg of *n*-3 (560 mg EPA + 2000 mg DHA) and observed a protective effect against TBI [59]. Together, these findings highlight the potential of *n*-3, particularly DHA, as a strategic intervention to enhance neuroprotection in high-risk populations such as SMs. By reducing neural damage and preserving cognitive function, *n*-3 could play a vital role in maintaining operational capabilities and improving long-term neurological health within the military.

In the realm of physical injury prevention, *n*-3 have demonstrated potential as both a biomarker for injury risk and a therapeutic tool for recovery. One study found that recreational runners with higher O3I levels were less likely to experience RRI, highlighting the possibility of utilizing *n*-3 status as a biomarker to assess injury risk, offering a practical approach to guide dietary interventions [62]. For SMs, whose physical demands often parallel or exceed those of athletes, monitoring O3I levels could enable targeted strategies to reduce injury rates, thereby enhancing overall force readiness. Expanding upon this, a different study identified lower O3I levels in individuals with degenerative rotator cuff tears compared to healthy controls, suggesting a potential link between *n*-3 inadequacies and tendon vulnerability [63]. This is particularly relevant for SMs, for whom tasks involving repetitive or heavy lifting place significant strain on musculoskeletal systems. While longitudinal research is needed to confirm causation, optimizing *n*-3 status may represent a proactive approach to reducing the risk of tendon-related injuries, preserving operational capacity.

Research into *n*-3 supplementation as a treatment for injuries further supports its role in musculoskeletal health. A study demonstrated that participants facing muscle disuse experienced significantly less muscle atrophy and higher muscle protein synthesis rates when supplemented with *n*-3 during immobilization [64]. For SMs recovering from injuries or surgeries, such supplementation could mitigate muscle loss and enhance the rehabilitation process, facilitating quicker returns to full physical activity and duty.

These findings underscore the value of incorporating *n*-3 into injury prevention and recovery protocols. By reducing injury risk, preserving muscle integrity during periods of inactivity, and supporting rehabilitation, *n*-3 offers a promising strategy to maintain physical readiness and enhance long-term health outcomes in physically demanding environments.

### 4.5. Strength and Power

As discussed earlier, two studies demonstrated associations between dietary *n*-3 intake and HGS, suggesting a potential role for supplemental *n*-3 in enhancing strength [59,67]. This could hold particular significance for SMs, where grip strength is often crucial for tasks such as carrying equipment and climbing.

One study further explored performance metrics, finding that EPA levels positively correlated with plank performance. Such findings suggest that *n*-3 may support muscular endurance—a key element in the physical fitness regimens of SMs [68]. A later study expanded on these findings, reporting improvements in jump performance, strength, and power metrics in groups receiving EPA and DHA compared to placebo [45]. These improvements could enhance physical readiness in operational settings, particularly where explosive power and agility are required. Similar benefits were reported by another study, which found that squat jump and countermovement jump performance improved more significantly in groups receiving higher EPA doses [52]. In professional rugby players, a study observed that *n*-3 helped maintain explosive power during pre-season training [48]. These results suggest potential applications for SMs engaged in rigorous training programs, enabling them to sustain peak performance levels over time. Further supporting this, research observed reduced Wingate percent power drop when subjects were provided *n*-3, indicative of improved neuromuscular function and the ability to sustain anaerobic power for longer durations [70].

The potential for *n*-3 to improve strength, power, and neuromuscular function is promising for SMs. These adaptations could enhance operational performance, improve recovery times, and contribute to overall physical readiness. Given the diverse and physically demanding nature of military tasks, optimizing *n*-3 intake may represent an important strategy for sustaining peak physical capabilities under a range of conditions.

### 4.6. Summary

The domains with the strongest evidence include cardiopulmonary function, exercise recovery, and cognitive function, while injury prevention and strength and power have emerging, yet more limited, evidence. In cardiopulmonary studies, certain research suggests potential benefits—such as improvements in stroke volume [21] and aerobic capacity [24,28,29]—while some other studies show no significant changes in heart rate, blood pressure, or pulmonary function [14,16,17,18,20,21,22,23,24,25,26,27,28,29,30,31,74]. Similarly, the impact of *n*-3 on exercise recovery—such as muscle soreness, CRP levels, and muscle damage markers—has yielded many positive results. However, some studies do not demonstrate consistent benefits [32,33,34,35,37,38,39,41,42,43,44,45,46,47,48,49,50,51,52,75]. In cognitive function, some studies report enhancements in attention and reactivity [53,54,55,56] and information processing [56,57], while others found no significant effects on executive function or overall cognitive performance [53,54,55,56,57,58]. Likewise, *n*-3 has produced some positive results on various performance outcomes [45,59,67,68,70], yet there are some discrepancies in measures such as neuromuscular adaptations and jump performance [39,45,48,52,59,67,68,70,71,72]. These mixed findings underscore the need for further research to fully elucidate the effect of *n*-3 on performance outcomes for SMs.

### 4.7. Strengths and Limitations

This review highlights several strengths of existing research on *n*-3. Studies included in this analysis were rigorously assessed for quality, ensuring the reliability of findings and adherence to high methodological standards. Notably, many studies employed randomized controlled trial designs, the gold standard in clinical research, to evaluate the efficacy of *n*-3 interventions. Additionally, the incorporation of biomarkers, such as O3I, provided quantifiable insights into the relationships among *n*-3 status, *n*-3 supplementation, and performance outcomes, enhancing the applicability of findings.

Despite these strengths, several limitations must be considered. A key challenge lies in the variability in dosages and durations across studies, which complicates cross-study comparisons and limits the establishment of standardized recommendations. Furthermore, the heterogeneity in study populations, including differences in age, sex, and baseline health, raises questions about the generalizability of findings, particularly to younger and more physically active groups such as SMs. Additionally, many studies examining *n*-3 and exercise have shown considerable variability in exercise type, duration, and intensity, making it difficult to draw consistent conclusions about the effect on *n*-3 on physical performance. Research on *n*-3 for SMs in military settings has faced challenges of low compliance, which should be considered when evaluating the effectiveness of *n*-3 in military settings. Issues with low compliance observed in a few studies highlight the need for exploring solutions and developing best practices for improving adherence to *n*-3 supplementation. Additionally, there is a lack of longitudinal evidence, as many of the studies were relatively short in duration, leaving the long-term impacts of *n*-3 uncertain. Longitudinal research is essential for understanding how sustained *n*-3 supplementation affects key health and performance outcomes over time. This could enable the development of more precise guidelines on dosage, duration, and target populations, particularly for military personnel with distinct physical and cognitive demands. Taken together, these limitations contribute to the variability in results across studies, with some reporting significant benefits and others finding minimal effects. Addressing these gaps through well-designed, long-term trials with standardized protocols will be critical for establishing clearer recommendations and maximizing the potential benefits of *n*-3 in the military and other tactical populations.

### 4.8. Future Research

Future research on *n*-3, particularly in tactical populations, should focus on determining the optimal status, dose, and type of *n*-3 (i.e., EPA, DHA, or a combination), if applicable, for various performance outcomes. While studies have demonstrated the potential benefits of *n*-3, specific recommendations for optimal performance remain unclear. To gather more consistent evidence across the various performance outcome measures, a study duration of 8–12 weeks assessing O3I from red blood cells is recommended. Standardizing both the measurement method and study duration would facilitate clearer associations between outcomes.

## 5. Conclusions

In conclusion, *n*-3 have shown potential benefits across various performance domains, with the most evidence supporting cardiopulmonary function, exercise recovery, and cognitive function. Other areas such as injury prevention and strength and power have limited evidence. Although some studies highlight positive outcomes related to *n*-3 status and supplementation, others fail to demonstrate significant or consistent effects. Bridging these gaps will be essential to establish targeted, evidence-based *n*-3 guidelines to optimize the performance of tactical populations, particularly SMs.

## Figures and Tables

**Figure 1 nutrients-17-00307-f001:**
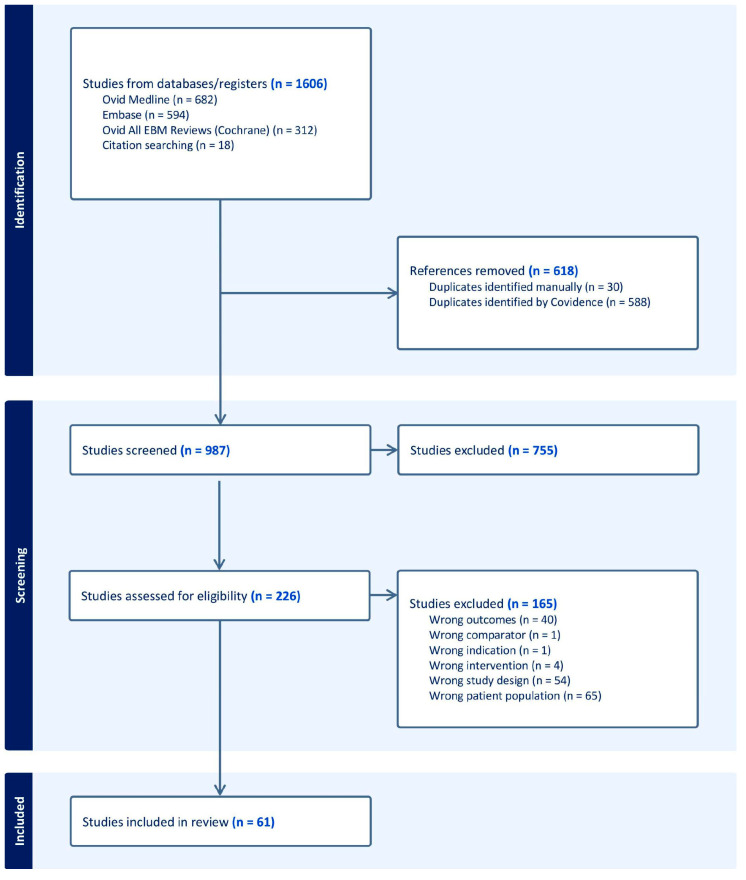
PRISMA flow chart outlining the article selection process.

## Data Availability

No new data were created or analyzed in this study. Data sharing is not applicable to this article.

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
