# Peer review of "Examining the Influence of Omega-3 Fatty Acids on Performance, Recovery, and Injury Management for Health Optimization: A Systematic Review Focused on Military Service Members"

_nutrients, 2025, doi:10.3390/nu17020307_

Round 1

Reviewer 1 Report

Comments and Suggestions for Authors

Thank you very much for this review and the presentation of results. However, the text is sometimes boring to read.  Why? You choose the presentation everytime like...There are some positive results......but there are some negative results. You choose the same presentation in the discussion, that is boring.l  The reader want to know what is right and what is wrong - we don't know it. Remember the 50 years old discussion about the impact of O3FA on cardiovascular mortality with thousands of probands. After each positive study, there follow a negative study sponsored by steakholder who are interesterested to push O3FA away from the market. Please analyze your data to find what is the reason for different results. For example we agree that O3FA reduce heart rate. Why some studies didn't find this fact: too low dose, patient selection, BMI, age!!, is the compliance proofed by blood levels, timeline of publications and which journal publish negative or positive results and most of all who is the sponsor of the authors.

Such an analysis would help the reader.

Author Response

Comment: Thank you very much for this review and the presentation of results. However, the text is sometimes boring to read.  Why? You choose the presentation everytime like...There are some positive results......but there are some negative results. You choose the same presentation in the discussion, that is boring. The reader want to know what is right and what is wrong - we don't know it. Remember the 50 years old discussion about the impact of O3FA on cardiovascular mortality with thousands of probands. After each positive study, there follow a negative study sponsored by stakeholders who are interested to push O3FA away from the market. Please analyze your data to find what is the reason for different results. For example we agree that O3FA reduce heart rate. Why some studies didn't find this fact: too low dose, patient selection, BMI, age!!, is the compliance proofed by blood levels, timeline of publications and which journal publish negative or positive results and most of all who is the sponsor of the authors.

Such an analysis would help the reader.

Response: Thank you for your suggestion. In the results section, we focus solely on reporting the outcomes of the studies without speculating on the reasons for differing findings. However, in the discussion section, we address potential factors contributing to the variability in results and suggest possible strategies to reduce this variability in future research. We expanded upon the factors contributing to varying results in lines 593–623.

Reviewer 2 Report

Comments and Suggestions for Authors

The reviewer thinks that this manuscript needs of rewriting of some parts in Introduction and Results 

Some specific comments and recommendations to the authors are noted below.

 The reviewer’s remarks are highlighted in orange in the attached file

Introduction

The reviewer recommends including a list of abbreviations used in all manuscript

Results

3.2.3. Muscle Damage Markers

it is good to shorten the very detailed description of the established findings in the articles

"another study" is repeated too often at the beginning of several sentences (row 246-261) and once again in row 284 - 310

Author Response

Comment: Introduction: The reviewer recommends including a list of abbreviations used in all manuscript

Response: Thank you for your recommendation. We have added a list of abbreviations used in the manuscript for ease of reference. We have attached it as supplemental material. 

Comment: Results: 3.2.3. Muscle Damage Markers, it is good to shorten the very detailed description of the established findings in the articles

Response: Thank you for your suggestion, we have removed some unnecessary details from the section to flow better. Please see lines 239-265 for the changes. 

Comment: "another study" is repeated too often at the beginning of several sentences (row 246-261) and once again in row 284 - 310

Response: Thank you for pointing this out. We have diversified the language used in this section. Please see lines 239-329 for the changes.

Reviewer 3 Report

Comments and Suggestions for Authors

A very well conducted systematic review introducing a comprehensive screening of existent literature regarding dietary supplementation in a less studied target population.

Abstract, introduction and description of the methods used are concise, informative and correctly introduce the motivation for this report. Figure 1 introducing the method flow chart could maybe read more intelligible.

This reviewer highly appreciates the display of results based upon key domains of n-3 impact on health

The  authors identify in their report  inconsistent and inconclusive evidence regarding the benefits of n-3 on several performance outcomes ( such as  strength, recovery, and injury prevention). A discussion could be expanded  on possible reasons for this variability and on the causes such obviously important (for this category) study outcomes are left unaddressed. Such approach could possible orient future work to address these less explored outcomes.

Regarding the markers of muscle damage (basically like many other situations involving a form a physical exercise) a large variability in exercise type , duration, intensity can be noted. This adds greatly to the already large variability of sources/doses of n-3. 

Have the studies identified gender variability in respect to outcome measures (physical exercise-wise impact but as well the other key domains discussed?) The study mentioned in R534 is singular or are they other studies focusing on gender specificity in regard to n-3 supplementation

The discussion could benefit from a deeper exploration of mixed results regarding cognition  especially for domains like executive function.

Regarding injury prevention  authors state n-3 potential in reducing neural and musculoskeletal injuries but acknowledges limited evidence resulting from current literature,  A critical discussion on why such evidence is scarce (maybe  logistical challenges in military settings or limitations in injury metrics) could potentially inform future investigations.

From the literature presented it is obvious there is a lack of longitudinal evidence, as the authors mention themselves. Maybe elaborating on why longitudinal research is necessary and how it could better inform guidelines for n-e supplementation in this special category might strengthen the argument for future research.

Several compliance issues are mentioned regarding n-3 administration. Exploring solutions or best practices for improving adherence to omega-3 supplementation could add practical value for this category but maybe for overall compliance in larger populations.

Regarding future research can the authors forward several possible suggestions for future research designs (including study duration and outcome measures)  that would  allow the collection of more uniform evidence?

Author Response

Comment 1: Abstract, introduction and description of the methods used are concise, informative and correctly introduce the motivation for this report. Figure 1 introducing the method flow chart could maybe be read more intelligible.

Response: Thank you for your feedback. To enhance the clarity of the sentence introducing the flow chart, we have revised. Please refer to lines 67–70 for the updated version.

Comment 2: The authors identify in their report  inconsistent and inconclusive evidence regarding the benefits of n-3 on several performance outcomes (such as  strength, recovery, and injury prevention). A discussion could be expanded  on possible reasons for this variability and on the causes such obviously important (for this category) study outcomes are left unaddressed. Such an approach could possibly orient future work to address these less explored outcomes.

Response: Thank you for your comment, we discuss possible reasons for the inconclusive evidence in the limitations section. To clarify we have added a section emphasizing these factors and highlighting future work. This section reads “Taken together, these limitations contribute to the variability in results across studies, with some reporting significant benefits and others finding minimal effects. Addressing these gaps through well-designed, long-term trials with standardized protocols will be critical for establishing clearer recommendations and maximizing the potential benefits of n-3 in the military and other tactical populations” Please see lines 593-614 for more detail. 

Comment 3: Regarding the markers of muscle damage (basically like many other situations involving a form of physical exercise) a large variability in exercise type, duration, intensity can be noted. This adds greatly to the already large variability of sources/doses of n-3. 

Response: Thank you for pointing this out. We agree this is a limitation to the research, and have added some more detail to the limitations section. The added section reads “​​Additionally, many studies examining n-3 and exercise have shown considerable variability in exercise type, duration and intensity, making it difficult to draw consistent conclusions about the effect on n-3 on physical performance.” Please see lines 598-600 for the changes. 

Comment 4: Have the studies identified gender variability in respect to outcome measures (physical exercise-wise impact but as well the other key domains discussed?) The study mentioned in R534 is singular or are they other studies focusing on gender specificity in regard to n-3 supplementation

Response: Thank you for pointing this out. Only one study specifically focused on women, while the remaining studies included both men and women without distinguishing results by gender. As a result, gender variability was not a primary focus of this review. Since we did not address gender-specific effects, we have replaced “young women” with “participants” to avoid any potential confusion regarding gender-related outcomes.

Comment 5: The discussion could benefit from a deeper exploration of mixed results regarding cognition  especially for domains like executive function.

Response: Thank you for this suggestion. We expanded on factors contributing to variability in results across all categories in lines 593–614. To expand upon executive function specifically, we have added details in lines 357-360. 

Comment 6: Regarding injury prevention  authors state n-3 potential in reducing neural and musculoskeletal injuries but acknowledge limited evidence resulting from current literature,  A critical discussion on why such evidence is scarce (maybe  logistical challenges in military settings or limitations in injury metrics) could potentially inform future investigations.

Response: This theme of limited evidence and current literature is true for all of the topics discussed in this review. We expanded upon this discussion in the limitations section, starting at line 593. 

Comment 7: From the literature presented it is obvious there is a lack of longitudinal evidence, as the authors mention themselves. Maybe elaborating on why longitudinal research is necessary and how it could better inform guidelines for n-e supplementation in this special category might strengthen the argument for future research.

Response: To address your comment, we have expanded on the importance of longitudinal research. The added section states: “Longitudinal research is essential for understanding how sustained n-3 supplementation impacts key health and performance outcomes over time. This could enable the development of more precise guidelines on dosage, duration, and target populations, particularly for military personnel with unique physical and cognitive demands.” Please refer to lines 606–610 for the revisions.

Comment 8: Several compliance issues are mentioned regarding n-3 administration. Exploring solutions or best practices for improving adherence to omega-3 supplementation could add practical value for this category but maybe for overall compliance in larger populations.

Response: Thank you for this comment. We have added some more detail to the conversation around low compliance. The added portion reads “Issues with low compliance observed in a few studies highlight the need for exploring solutions and developing best practices for improving adherence to n-3 supplementation.” Please see lines 603-604 for the changes. 

Comment 9: Regarding future research, can the authors forward several possible suggestions for future research designs (including study duration and outcome measures)  that would  allow the collection of more uniform evidence?

Response: Thank you for this suggestion. In the future research section we have added a couple recommendations that would allow for a collection of more uniform evidence. This section reads “To gather more consistent evidence across various performance outcome measures, a study duration of 8-12 weeks assessing O3I from red blood cells is recommended. Standardizing both the measurement method and study duration would facilitate clearer associations between outcomes.” Please see lines 620-623 for the changes.

Reviewer 4 Report

Comments and Suggestions for Authors

The manuscript addresses the effect of n3 fatty acids on different kinds of health and nutritional aspects in human diet. I think it is interesting, novel, and provides a wide information. Some performances ought to be done before its acceptation.

Title

It could be modified. Studies presented in the Review are not especially focused on military service members, but on people in general.

Abstract

Around 280 words are included. I think it is somewhat long.

Results and Discussion sections

Both could be joined and spare space. This is a Review, not an experimental study.

General

Very short comments are provided on chemical and nutritional aspects of the different kinds of foods and their content on n3 fatty acids. I think the review fails in the sense of recommending the intake of concrete foods (i.e., seafoods, especially high-lipid content seafoods). N3 fatty acids are treated as a relatively abstract concept. I understand the paper has a medical/clinical focus, but chemical/nutritional aspects are on the basis of the importance of n3 fatty acids.

Author Response

Title

Comment 1: It could be modified. Studies presented in the Review are not especially focused on military service members, but on people in general.

Response: Thank you for your suggestion. We have updated our title to “Translating Evidence on Omega-3 Fatty Acids for Military Performance, Recovery, and Injury Management: A Systematic Review” 

Abstract

Comment 2: Around 280 words are included. I think it is somewhat long.

Response: Thank you for pointing this out. We have shortened the abstract to 250 words. 

Results and Discussion sections

Comment 3: Both could be joined and spare space. This is a Review, not an experimental study.

Response:  Thank you for your suggestion. Given the length of these sections, we have separated them to improve readability. The results solely state the study outcomes while the discussion section offers context on the mixed results and highlights the potential benefits of O3I. 

General

Comment 4: Very short comments are provided on chemical and nutritional aspects of the different kinds of foods and their content on n3 fatty acids. I think the review fails in the sense of recommending the intake of concrete foods (i.e., seafoods, especially high-lipid content seafoods). N3 fatty acids are treated as a relatively abstract concept. I understand the paper has a medical/clinical focus, but chemical/nutritional aspects are on the basis of the importance of n3 fatty acids.

Response:  Thank you for your feedback. Although this paper does not focus extensively on the chemical or nutritional details, we introduce these concepts in the introduction to provide a foundational understanding, including the different types of omega-3s and their various sources. For more information on omega-3-rich food intake, please refer to lines 45–56. To add more detail on the chemical aspect of n-3, we have added “long chain fatty acids” to the description on EPA and DHA in line 43.

Round 2

Reviewer 1 Report

Comments and Suggestions for Authors

The manuscript is ok

Author Response

Comment: The manuscript is ok

Response: Thank you for your review and confirmation. We appreciate your time and approval of our manuscript.

Reviewer 4 Report

Comments and Suggestions for Authors

The manuscript has been modified in some aspects previously mentioned. However, other concerns remain as in the previous version.

Title

As previously mentioned, I think it ought to be modified. The content of the manuscript is a review which includes a wide range of studies on human health, but not especially on military service members. I would propose something like: Examining the influence of omega-3 fatty acids on performance, recovery, and injury management for human health: A systematic review focused on military service members.

General

Very scarce comments have been included regarding the chemical and nutritional aspects of the different kinds of foods and their content on n3 fatty acids. As previously mentioned, n3 fatty acids are treated as a relatively abstract concept. I think the review still fails in the sense of recommending the intake of concrete foods (i.e., seafoods, especially those including high-lipid contents).

Author Response

Comment 1: As previously mentioned, I think it ought to be modified. The content of the manuscript is a review which includes a wide range of studies on human health, but not especially on military service members. I would propose something like: Examining the influence of omega-3 fatty acids on performance, recovery, and injury management for human health: A systematic review focused on military service members.

Response 1: Thank you for your suggestion. We have updated our title to “Examining the Influence of Omega-3 Fatty Acids on Performance, Recovery, and Injury Management for Health Optimization: A Systematic Review Focused on Military Service Members.”  

Comment 2: Very short comments are provided on chemical and nutritional aspects of the different kinds of foods and their content on n3 fatty acids. I think the review fails in the sense of recommending the intake of concrete foods (i.e., seafoods, especially high-lipid content seafoods). 

Response 2:  Thank you for your feedback, we feel we have included the foundational aspects of  omega-3 fatty acids that you are looking for in the introduction. We introduce omega-3 fatty acids to provide a foundational understanding, including the different types of omega-3s and their various sources. Details on omega-3-rich food intake was added in lines 45–56. We list foods high in omega-3 fatty acids (lines 43-44) and provide the recommended amount of 2-3 servings per day as a reference (lines 56-57).